# ‘Get a Fish’ vs. ‘Get a Fishing Skill’: Farmers’ Preferred Compensation Methods to Control Agricultural Nonpoint Source Pollution

**DOI:** 10.3390/ijerph17072484

**Published:** 2020-04-05

**Authors:** Xiaoping Li, Yan Yan, Liuyang Yao

**Affiliations:** 1School of Public Administration and Law, Chang’an University, Xi’an 710064, China; lxp11742@163.com; 2College of Economics and Management, Northwest A & F University, Yangling 712100, China; yyan@nwafu.edu.cn; 3International Business School, Shaanxi Normal University, Xi’an 710119, China

**Keywords:** farmers’ preference, compensation method, agricultural non-point source pollution, multivariate probit model

## Abstract

Ecological compensation is an important means for controlling agricultural nonpoint source pollution, and compensation methods comprise an essential part of the compensation policy for mitigating this form of pollution. Farmers’ choice of compensation methods affects their response to compensation policies as well as the effects of pollution control and ecological compensation efficiency. This study divides ecological compensation methods into two distinct philosophies—the “get a fish” method (GFM) and “get a fishing skill” method (GFSM)—based on policy objectives, to determine farmers’ choice between the two methods and the factors influencing this choice. Furthermore, by analyzing survey data of 632 farmers in the Ankang and Hanzhong cities in China and using the multivariate probit model, the study determines farmers’ preferred option among four specific compensation modes of GFM and GFSM. The three main results are as follows. (1) The probability of farmers choosing GFM is 82%, while that of choosing GFSM is 51%. Therefore, GFM should receive more attention in compensation policies relating to agricultural nonpoint source pollution control. (2) Of the four compensation modes, the study finds a substitution effect between farmers’ choice of capital and technology compensations, capital and project compensations, material and project compensations, while there is a complementary relationship between the choice of material and technology compensations. Therefore, when constructing the compensation policy basket, attention should be given to achieving an organic combination of different compensation methods. (3) Highly educated, young, and male farmers with lower part-time employment, large cultivated land, and a high level of eco-friendly technology adoption and policy understanding are more likely to choose GFSM. Hence, the government should prioritize promoting GFSM for farmers with these characteristics, thereby creating a demonstration effect to encourage transition from GFM to GFSM.

## 1. Introduction

Countries worldwide are facing serious agricultural nonpoint source pollution [1,2,3]. Excessive chemical fertilizers, pesticides, livestock manure in agricultural production have led to N, P, and other pollutants entering the water and the entire ecosystem, manifesting in the form of agricultural nonpoint source pollution [4]. Agricultural nonpoint source pollution depletes the water quality, causes ecological imbalance, and threatens food safety and human health. According to the Best Management Practices (BMPs), this type of pollution can be controlled through engineering and non-engineering methods [5,6,7,8,9]. The engineering methods mainly include the construction of plant buffer zones, restoration of farmland to forest, and implementation of a fallow land policy [6]. Non-engineering means mainly include the development of environmental regulations, fertilizer and nutrient management systems, and integrated pest management techniques [7,8]. Although these methods have achieved positive outcomes, they have been limited by certain issues [10,11]. The most prominent challenge is that farmers lack the willingness to voluntarily participate in these practices [12]. To achieve incentives, economists suggest that the ecological compensation can equip farmers to control agricultural nonpoint source pollution voluntarily [13,14].

Farmers’ choice of ecological compensation policies affects the compensation efficiency and pollution control effects [15,16]. Scholars have conducted extensive research on how to improve farmers’ enthusiasm to participate in ecological compensation policies. At present, relevant studies mainly analyze farmers’ willingness to participate [17] and calculate reasonable compensation standards [18,19,20,21]. However, few studies discuss how to execute the compensation methods available to farmers.

The mode of ecological compensation determines how compensation will reach farmers [22]. According to policy objectives and their effects, this study divides ecological compensation methods into two categories—the get a fish method (GFM) and get a fishing skill method (GFSM). These two categories comprise two modes each; the former comprises capital and material compensations, and the latter comprises technology and project compensations. In practice, subject to transaction costs and operational convenience, government-led ecological compensation is often based on GFM [14,23]. This way of directly distributing capital or material to the ecological protectors is often accompanied by defects of persistent weakness, low compensation standards, and the inability to prevent moral hazard [19], and has therefore been questioned by many scholars [24].

At the same time, many scholars have recognized the role of GFSM in improving regional self-development and enabling eco-protectors to generate sustainable income. For example, the book “Environmental Capital Operation—Unification of Economic Benefits and Ecological Benefits” [25] expounded how environmental protection entrepreneurs achieved the dual goals of environmental protection and profit creation through GFSM such as leisure agriculture, ecotourism, and eco-friendly real estate in the early 20th century. Similarly, Meng et al. [26] considered GFSM as the key to source-water protection. Zhao and Wang [27] revealed that GFSM is crucial to closing a significant gap in compensation funds and resolving the difficulty in quantifying the amount of compensation. Wang et al. [28] suggested that GFSM can improve the self-development awareness and ability of forest farmers and contribute toward realizing the sustainable development of the economy, society, and ecology in forest communities. These studies mostly analyze ecological compensation methods at a theoretical level. In other words, they lack an empirical analysis of compensation methods, especially ignoring recipients’ preference and response to compensation methods. A compensation method can achieve a better incentive effect if it is designed according to farmers’ preference.

In the process of implementing the ecological compensation policy for agricultural nonpoint source pollution control, a misalignment between the compensation method and farmers’ preference will directly affect farmers’ pollution control behaviors, making the compensation policy inefficient and ineffective. In this regard, this study aims to answer the following two questions: Which should be the leading ecological compensation method for agricultural nonpoint source pollution control—GFM or GFSM? What factors affect farmers’ preference for those compensation methods? Considering that the ecological compensation method policy set out may be an organic combination of multiple compensation modes of GFM and GFSM, this study uses the multivariate probit model to analyze farmers’ preference for these modes and the factors influencing their choice. A clarification on these critical issues will help us to build a set of compensation policies that meet the values and interests of farmers as well and encourage farmers to protect the environment. The rest of this paper is arranged as follows. Section 2 introduces the materials and methods, including a comparison between the four compensation methods, the research area, the research design, and the mathematical model. Section 3 analyzes farmers’ preference and its influencing factors. Section 4 provides conclusions and policy implications.

## 2. Materials and Methods

### 2.1. Two Types of Compensation: GFM and GFSM

There are significant differences between GFM and GFSM. GFM is concerned with directly distributing money or goods to the eco-protectors to compensate for the lost opportunity cost as a result of implementing eco-environmental protection measures [24]. It aims to guarantee the basic livelihood of eco-protectors, thereby allowing a quick short-term compensation effect with a slight risk [29]. However, GFM has the following two shortcomings. First, it cannot be organically integrated with the economic system because it lacks sustainability. Once a compensation project is completed, the individual receiving compensation will return to the previous production patterns characterized by high pollution and consumption [19]. Second, GFM can easily lead to compensation dependence, driving recipients to exhibit “lazy behavior” [29]. GFSM mainly includes technical training, project investment, and industrial support. It aims to improve the self-development awareness and income-generating ability of recipients. The advantage of GFSM is its sustainability. It can improve the sustained income-generating ability of recipients and improve the coordination of the ecological-socioeconomic system. However, GFSM is also limited by its slow effect, and hence long realization time [25,30].

Specifically, there are four main modes of providing ecological compensation—capital, material, technology, and project compensations. The former two are categorized under GFM, while the latter two are categorized under GFSM owing to their ability to improve the reproduction capacity of farmers. To reduce the hypothetical bias, the first step in the survey aims to show farmers the advantages and disadvantages of different compensation methods, as shown in Table 1.

### 2.2. Research Areas

Ankang and Hanzhong cities in the Shaanxi Province of China are selected as the research areas (Figure 1). Shaanxi Province is located in central China and the upper reaches of the Yangtze River. It is a major agricultural province in China. Ankang and Hanzhong cities are situated in the southern part of Shaanxi Province. The main crops of these cities are rice, rapeseed, wheat, and maize, and the grain crops are double-cropped. These two cities are rich in natural resources and play a key role in the construction of the ecological civilization in China. China has chosen Ankang City as *the pilot demonstration city for the construction of national main functional areas* and the *National Forest City*, while Hanzhong City has been recognized as the *National Historic and Cultural City*, the *National Ecological Construction Demonstration City*, and the *National Garden City*. With respect to the demographics and agricultural statistics, in 2016, the total population of Ankang City stood at 264,000; the traditionally cultivated land area accounted for 197,300 ha; and the total agricultural output value was 6.09 billion yuan. During the same period, the total population of Hanzhong City was 343,000; the area of traditionally cultivated land was 205,150 ha; and the total agricultural output value was 11.86 billion yuan. As for the nonpoint pollution sources in the cities, Ankang City experiences an annual runoff of 10.7 billion m^3^, with 73 rivers having a catchment area of more than 100 km^2^. In Hanzhong City, the mainstream of the Hanjiang River is 277.8 km, which accounts for 18.1% of the total length of the Hanjiang River; the basin area of this mainstream is 19,692 km^2^, which accounts for 72.3% of the total land area (27,246 km^2^) of Hanzhong City.

There are four main reasons these cities were chosen as the study areas. First, agricultural nonpoint source pollution became the primary source of water pollution in the study area [31,32]. For example, owing to water pollution, the Hanjiang River carries 1.7 mg/L of nitrogen when it flows out of Ankang City (Hanzhong City is upstream) [33], which can be classified into four types of water (I–IV) according to the *China environmental quality standard for surface water*. Second, those areas experience excessive fertilizer application. According to the survey data of agricultural production, the inputs of chemical fertilizers for rice, wheat, and rapeseed were 328.35 kg/hm^2^, 297.15 kg/hm^2^, and 319.65 kg/hm^2^, respectively. After considering the characteristics of the double-cropped main crops in the study area (rice, wheat, and rapeseed, respectively), the annual average input of chemical fertilizers per hectare ranged from 625.50 kg to 648.00 kg, which was 2.78–2.88 times the internationally recognized upper limit of fertilizer application (225 kg/hm^2^). Therefore, the ecological compensation policy in this region holds significance in view of controlling agricultural nonpoint source pollution and ensuring ecological security. Third, these cities are critical water sources for China, so there are many beneficiaries of its agricultural nonpoint source pollution control. They house the headwaters of the Hanjiang River as well as the water conservation areas of the South-to-North Water Transfer Project. Hanjiang River is the largest tributary of the Yangtze River, which is the largest river in China. The South-to-North Water Transfer Project is the largest water conservancy project in China. Once water pollution occurs in this area, the pollutants may spread across the Hanjiang River, even endangering Beijing-Tianjin and the Yangtze River basin. Fourth, the control of agricultural nonpoint source pollution in the study area is of great ecological significance. These cities are located in the hinterland of the Qinba Biodiversity Ecological Function Zone, which is a crucial national ecological function area. Hence, measures to control agricultural nonpoint source pollution in the study areas contribute immensely toward the safety of the ecological function area and that of the wildlife habitats.

### 2.3. Research Design

The study’s data are taken from two questionnaire surveys conducted by a research group in the study areas in December 2016 and December 2017. During the entire data collection period, there was no change in policy to address the ecological condition or in relation to rice production. First, considering the socioeconomic conditions, such as rice-planting area, economic development level, and population proportion, in the study areas, the Hanyin County, the Hanbin District, and the Pingli County were selected as survey areas in Ankang City; the Mianxian County and the Chenggu County were selected as survey areas in Hanzhong City (as shown in Figure 1). Subsequently, two to three natural villages were randomly selected from each county. In each village, 50 to 60 farmers were randomly selected for two face-to-face questionnaire surveys. Of the 670 questionnaires sent for the surveys, 632 valid questionnaires were collected. The validity rate of the questionnaires was 94.33%. The questionnaire mainly comprised questions on the individual and family characteristics, agricultural management characteristics, psychological awareness, and policy environment of the respondents. The endowment characteristics, such as household income and agricultural production, were taken from the data of 2016.

To reduce the temptation bias, we trained the investigators and asked them to describe the questionnaire objectively. To reduce strategic bias, the investigators showed research and identity certificates to respondents and emphasized the academic purpose of the research. To reduce deviation from survey methods, the survey was conducted in the form of face-to-face interviews.

### 2.4. Mathematical Model

In practice, the compensation policy may represent a combination of several compensation modes. Since farmers may choose one or more ecological compensation policies at the same time, this study uses the multivariate probit model to analyze farmers’ preference for ecological compensation. The advantage of this model is that it examines the correlation between multiple decisions and analyzes the influencing factors of the actors making multiple decisions at the same time [34,35].

If choosing the *j_th_* item of compensation mode, the farmer *i* will obtain the utility of *U_ij_*. The *ε_ij_* is the random errorterm, representing the influence of unobservable factors on the farmer’s utility. According to the stochastic utility theory, farmers will choose a compensation mode that will maximize their own welfare. Then, the multivariate probit model of farmers’ choice of compensation mode can be expressed as follows: (1)$$U_{ij}=\alpha_{j}+\underset{k}{\sum }\beta_{jk}X_{jk}+\varepsilon_{ij}$$
(2)$$Y_{ij}=\{\begin{array}{ccc} 1 & if\;U_{ij}\geq U_{in} & \forall j,\;n\in D,\;n\neq j\\ 0 & if\;U_{ij}<U_{in} & \forall j,\;n\in D,\;n\neq j \end{array}$$

In Formulas (1) and (2), *j* may take 1, 2, 3, and 4 to represent capital, material, technology, and project compensations, respectively. *U_ij_* is an unobservable utility variable. *Y_ij_* is the result of farmers’ choice. If the utility *U_ij_* of farmers’ choice of the *j_th_* compensation mode is greater than or equal to the utility *U_in_* of choosing another arbitrary compensation mode, then *Y_ij_* = 1, which means that farmers prefer this ecological compensation mode. *X_jk_* represents the socioeconomic variables affecting farmers’ ecological compensation mode. *β_jk_* is the corresponding estimation coefficient. The random error term *β_jk_* obeys the multivariate normal distribution with a mean value of 0. The covariance is *Ψ*, and then there is *ε_i_* = (*ε_i1_*,……,*ε_ij_*)~MVN[0,Ψ]. The covariance matrix is expressed as:(3)$$\Psi =\left(\begin{array}{cccc} 1 & \rho_{12} & \rho_{13} & \rho_{14}\\ \rho_{21} & 1 & \rho_{23} & \rho_{24}\\ \rho_{31} & \rho_{32} & 1 & \rho_{34}\\ \rho_{41} & \rho_{42} & \rho_{43} & 1 \end{array}\right)$$

In Formula (3), the non-diagonal elements represent unobservable relationships among the stochastic perturbation terms between four utility equations of farmers’ choice for four ecological compensation modes. The non-diagonal *ρ* value is not 0, which indicates a correlation between the random perturbation terms. In this case, the multivariate probit model should be used to estimate the model. When *ρ* > 0, it indicates that farmers’ choice of different ecological compensation modes has a complementary effect. When *ρ* < 0, it indicates a substitution effect.

## 3. Results

### 3.1. Variable Description

In this study, the dependent variable is denoted by farmers’ preference for the four modes of ecological compensation. We present the following scenario question to the farmers: In the agricultural production process, if the government encourages you to use organic fertilizer, microbial pesticides, and insecticide lamp instead of chemical, fertilizer and pesticide, which way or how would you like the government to compensate you for the possible economic losses? The answer options are as follows: A = capital; B = organic fertilizer, microbial pesticides, insecticide lamp, and other production materials; C = training on agricultural production technology; and D = government investment in the construction of environmental protection industries in the region. The above options correspond to capital, material, technology, and project compensations, among which farmers can choose one or more ecological compensation methods.

In terms of individual ecological compensation, of the 632 sample farmers, 414 (66%), 331 (52%), 273 (43%), and 108 (17%) farmers chose capital, material, technology and project compensation, respectively. The specific statistical results are shown in Figure 2.

Out of the sampled farmers, 268, 70, and 252 farmers chose only GFM, only GFSM, and both methods, respectively. However, 42 farmers did not choose either method. Among them, 520 (82%) and 322 (51%) farmers chose at least one kind of GFM and at least one kind of GFSM, respectively. The statistical results show that the probability of the sampled farmers choosing GFM is significantly higher than that of GFSM.

With respect to independent variables, several factors affect farmers’ attitudes and behaviors toward environmental protection. In this regard, some scholars provide evidence that respondents’ personal characteristics, family economic status, agricultural management characteristics [36], cognitive and policy environmental variables [37] are suitable for explaining farmers’ environmental protection behavior. These variables can improve the interpretation and prediction ability of the model [38,39]. Therefore, this study chooses individual, family, and agricultural management characteristics; cognitive variables; and policy variables to examine farmers’ preference for compensation methods.

Specifically, first, the individual characteristic variables are gender, age, and education degree of the respondents. Second, the family characteristic variables are household income, the population burden rate, and the degree of part-time farming. In this study, the population burden rate depicts the proportion of the aged members and children in the sampled families. The degree of part-time farming refers to the proportion of farmers’ income from non-agricultural industries to their total household income. In line with previous studies [40], when the proportion of non-agricultural income to total household income is [0, 0.1], it is used to refer to a pure farmer. When the proportion of non-agricultural income is (0.1, 0.5), it is used to define a type-I part-time household. When the proportion of non-agricultural income is (0.5, 1), it is used to define a type-II part-time household. Third, the agricultural management characteristic variables comprise the cultivated area and the degree of eco-friendly technology adoption. Fourth, the cognitive variables include farmers’ risk attitude and cognition of land’s ecological function. Risk attitude indicates the degree of an individual’s preference for risk [41]. In this study, farmers’ risk attitudes were measured by estimating the proportion of yield reduction caused by the total non-use of chemical fertilizers and pesticides in rice cultivation. The research team estimated the production yield of organic rice (using organic fertilizers and bio-pesticides instead of traditional fertilizers and pesticides) in the study areas. It was found that the yield reduction ratio of organic rice was about 0.4–0.6 than that of traditional rice. According to this phenomenon, when the estimated yield reduction ratios are (0, 0.4), (0.4, 0.6), and (0.6, 1), farmers’ preferences are categorized as risk-seeking, risk-neutral, and risk-averse, respectively. To measure farmers’ cognition of cultivated land’s ecological function, this study set the following question scenario: Do you think the cultivated land promotes water conservation, soil and water conservation, and biodiversity conservation? Finally, the policy variables focus on farmers’ understanding of the government’s policies for nonpoint source pollution control and related ecological compensation policies. This article chooses the following two questions to measure these variables: Have you heard about the government’s policies and measures for the comprehensive prevention and mitigation of agricultural pollution (control of fertilizers, pesticides, and agricultural waste)? and Have you heard about ecological compensation policies such as the restoration of farmland to forestry and the grassland or fallow policy? 

The statistical characteristics of each variable are shown in Table 2.

### 3.2. Estimated Results

This study used Stata 14.0 software to estimate the multivariate probit model. The covariance matrix of the regression equation is shown in Table 3. Statistical results show that the chi^2^-value of the model is 61.59, which passes the significant test at 1% of the statistical level. It shows correlations among the random disturbance items of each equation, implying mutual influences among different ecological compensation modes chosen by farmers. Hence, it is appropriate to use the multivariate probit model to analyze farmers’ preference for ecological compensation modes.

In the covariance matrix, the covariance coefficient of material and technology compensations is positive. This suggests the existence of a significant complementary effect between material and technology compensations. This result is consistent with the actual agricultural production scenario, in where most farmers do not use eco-friendly production materials. Therefore, farmers need technical guidance in the process of adopting the environmental protection materials. The covariance coefficient between capital and technology compensations as well as that between capital and project compensations is negative. This suggests that the probability of choosing GFSM is relatively small for farmers choosing capital compensation; conversely, the probability of choosing capital compensation is also relatively small for farmers choosing GFSM. The covariance coefficient between material and project compensations is negative. This suggests that farmers who choose material compensation are less likely to choose project compensation, conversely, the same. These conclusions further indicate apparent differences in farmers’ preference for GFM and GFSM. When designing the policy set of ecological compensation methods, it is crucial to consider the collocation of different compensation methods.

Table 4 shows the regression results of farmers’ preference for ecological compensation. The value of chi^2^ passed the significance test at the 1% statistical level, which shows that the fitting degree of the model notably. This means that the independent variables have strong explanatory power to the dependent variables.

This section discusses the influence of individual characteristic variables of farmers. Gender has a significant negative impact on farmers’ choice of capital and material compensations and a significant positive impact on project compensation, respectively. This shows that, compared to females, male farmers have a lower probability of choosing GFM, but a higher probability of choosing GFSM. The reason is that male farmers have higher acceptance of new technologies and policies [42]. Age exerts a significant positive effect on farmers’ choice of material compensation and a negative effect on project compensation. This means that older farmers are more likely to choose GFM. The reason is that elderly farmers have a low earning capacity, and hence they tend to choose GFM with stable income. Degree of education has a significant positive impact on farmers’ choice of technology and project compensations. It indicates that farmers with higher degree of education are more likely to choose GFSM. In general, educational experience can improve an individual’s comprehensive ability, and hence educated farmers are expected to have a better understanding of the ability of GFSM to increase income and protect the environment.

Among the family characteristic variables, household income has significant negative and positive impacts on farmers’ choice of material compensation and project compensation, respectively. In other words, the higher the household income, the lower the probability that the farmer chooses GFM will be, and the higher the probability of choosing GFSM will be. The reason for this may be that farmers with low income value financial and material support; however, high-income farmers tend to be more economically conscious. Degree of part-time farming has a significant positive impact on farmers’ choice of capital compensation, and negative impact of technology and project compensations. This means that farmers with a high degree of part-time farming express a higher willingness to choose GFM over GFSM. The reason is that such farmers tend to have a relatively stable non-agricultural income and incur a relatively high opportunity cost of farming.

Concerning agricultural management characteristic variables, cultivated land area has significant negative impact on farmers’ choice of capital compensation, and positive impact of project compensation. This shows that the larger the cultivated land area, the greater opportunity of farmers to choose GFSM and the smaller opportunity to choose GFM. The reason may be that the larger the cultivated land area, the more labor is consumed. Owing to a relatively low income from agriculture, laborers seek opportunities to improve production capacity. Eco-friendly technology adoption has significant negative and positive impacts on farmers’ choice of capital compensation and that of technology compensation, respectively. This indicates that the higher the adoption of eco-friendly technologies, the greater the probability that farmers choose GFSM will be, and the lower the probability of choosing GFM will be. The reason is that farmers may have previously gained from the adoption of more eco-friendly technologies. Additionally, they also gain a better understanding of the role of technical training in “saving and increasing income”, and hence they express a higher willingness to accept GFSM.

Concerning the impact of psychological cognitive variables, risk attitude has a significant positive impact on farmers’ choice of capital and project compensations. This means that farmers with a higher risk aversion express a higher willingness to choose GFM and GFSM, because they are less willing to take risks and more willing to accept all kinds of government support. Cognition of land ecological function has a significant negative impact on farmers’ choice of capital compensation and a positive impact on technology and project compensations. This indicates that farmers with a high cognition of the land’s ecological function would express a high willingness to choose GFSM and low willingness to choose GFM. Farmers’ understanding of the ecological function leads them to view farmland protection as a long-term project, and leads them to choose GFSM with sustainable characteristics.

Concerning the impact of the policy environment variables, understanding of the government’s nonpoint source pollution control policies has a significant negative impact on farmers’ choice of capital and material compensations and a positive impact of project compensation. This shows that, compared to GFM, farmers with a high awareness of cultivated land pollution control policies express a higher willingness to choose GFSM. Farmers who have a better awareness of the advantages and disadvantages of various governance measures are clearer of the advantages of GFSM versus GFM. Understanding of the ecological compensation policy has a significant positive impact on farmers’ choice of project compensation methods. It indicates that the higher the understanding of ecological compensation policy, the higher will be the willingness of farmers to choose GFSM. The reason may be that the higher the farmers’ understanding of the ecological compensation policy, the clearer will be their understanding of the need for long-term ecological protection.

## 4. Conclusions

The key to guaranteeing the effect of agricultural nonpoint source pollution control and the efficiency of compensation is to design compensation methods that meet farmers’ preference. Based on the survey data of 632 farmers in Ankang and Hanzhong cities, this study uses the multivariate probit model to analyze their choice of four specific compensation modes of GFSM and GFM. The main conclusions are as follows. First, the probability of farmers choosing GFM is 82% and that of choosing GFSM is 51%. The former is significantly higher than the latter, indicating that farmers prefer GFM. Second, there is a substitution effect between farmers’ choices of capital and technology compensations, capital and project compensations, material and project compensations. There is a complementary effect between that of material and technology compensations. Third, there are differences between the influences of independent variables on farmers’ choice for ecological compensation methods. The following variables exert significant influence on farmers’ preference for GFM: Gender, age, household income, the degree of part-time farming, cultivated area, the degree of eco-friendly technology adoption, the risk attitude, the cognition of land’s ecological function, and an understanding of government’s nonpoint pollution control policies. The following variables exert a significant influence on farmers’ preference for GFSM: gender, age, education degree, household income, the degree of part-time farming, cultivated area, the degree of eco-friendly technology adoption, the risk attitude, the cognition of the land’s ecological function, an understanding of government’s nonpoint source pollution control policies, and an understanding of ecological compensation policies.

Based on the above conclusions, this study mainly has the following policy implications. First, although many scholars advocate GFSM, specific scenario associated with the method should be analyzed. The results show that farmers prefer GFM currently, and hence the policy basket of ecological compensation should still focus on GFM as the main method and GFSM as the supplementary method. Second, attention should be given to matching different compensation modes when constructing the policy basket of ecological compensation modes for controlling agricultural nonpoint source pollution. For example, material compensation should be organically combined with technology compensation; in other words, technical training should be launched for promoting green production modes, such as organic fertilizer, green pesticides, and pesticide lamp. Capital compensation should not be placed along with technology and project compensations, while material compensation should not be placed along with project compensation; this matching would avoid policy mismatching. Third, in order to realize sustainable development of ecology, economy, and society, GFM should be gradually transformed into GFSM in the future. The government should prioritize promoting GFSM among young, male farmers with a high education level, a low degree of part-time farming, a large area of cultivated land, a high adoption degree of eco-friendly technologies, a high cognition of land’s ecological function, and a high degree of policy understanding. This will lead to demonstration and ripple effects. In the process of promoting GFSM, the government should improve farmers’ acceptance, at least, from the following two aspects. On the one hand, publicity can improve farmers’ awareness that GFSM has its sustainable and economic advantages. At the same time, technical training of environmental-friendly protection production can improve farmers’ ability to participate in GFSM. On the other hand, local economic conditions, employment of farmers, and local resources should be well considered during the process of specific GFSM design and implementation to ensure the compensation implement successfully.

In the follow-up study of compensation, it is necessary to explore temporal and spatial heterogeneity. In the long run, farmers’ preference for compensation mode may change with the development of policy, environment and personal ability. Understanding the internal mechanism of this change helps to improve the sustainability and effectiveness of compensation policy. From the spatial dimension, if the costs and benefits of the upper, middle and lower reaches are heterogeneous, differential compensation policies are indispensable. Discussion on how to optimize the distribution of the compensation in the Hanjiang River Basin is an important content of compensation research in the future.

## Figures and Tables

**Figure 1 ijerph-17-02484-f001:**
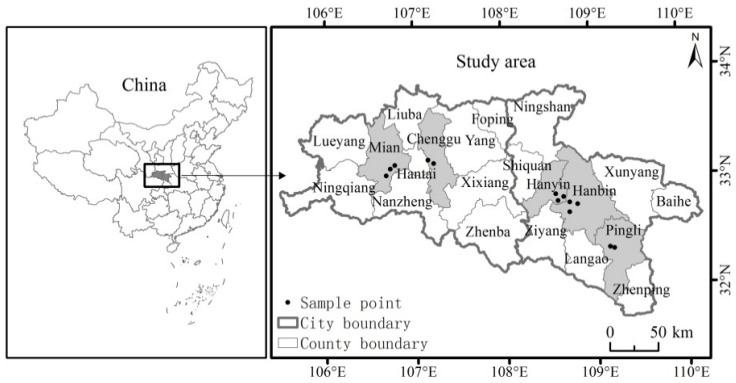
Research area and sampling point distribution.

**Figure 2 ijerph-17-02484-f002:**
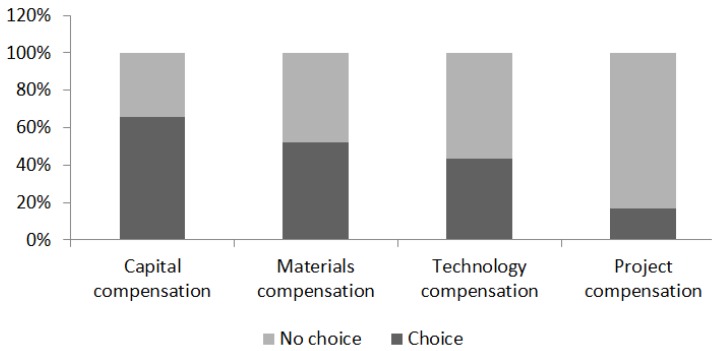
Farmers’ choice of four different ecological compensation methods.

**Table 1 ijerph-17-02484-t001:** Characteristics of GFM and GFSM.

Two Types of Compensation	Get a Fish Method (GFM)		Get a Fishing Skill Method (GFSM)	
Specific Method	Capital	Material	Technology	Project
Ccontent	Capital compensation or tax relief	Food, seeds, microbial pesticides, and even house	Technical guidance and related consulting services	Investing environmental industry
Advantage	work quickly	Guarantee productive capacity	Improve productivity	Promote recipients’ employment
Disadvantage	Capital misuse	Mismatch diversified demand	High time cost	Less consideration of f recipients’ opinions

**Table 2 ijerph-17-02484-t002:** Sample data descriptive statistics.

Variables	Assignment	Mean	Standard Deviation
**Individual Characteristics**			
Gender	Male = 1; female = 0	0.7120	0.4532
Age	Age	57.3813	10.2502
Education degree	Number of years	6.0759	3.7666
**Family Characteristics**			
Household income	[0, ¥20,000) = 1; [¥20,000, ¥40,000) = 2; [¥40,000, ¥60,000) =3; [¥60,000, ¥80,000) = 4; [¥80,000, ¥100,000) = 5; [¥100,000, +∞) = 6	3.0775	1.6848
Population burden rate	Population	0.3097	0.2443
Degree of part-time farming	Pure household = 1; type- I = 2; type- II = 3	1.9051	0.8448
**Agricultural Management Characteristics**			
Cultivated area	Cultivated area	4.1495	4.2757
Eco-friendly technology adoption	number of formula fertilization, organic fertilizer, and bio-pesticides	0.5364	0.7288
**Cognitive Variables**			
Risk attitude	Risk-seeking = 1; risk-neutral = 2; risk-aversion = 3	2.1203	0.8225
Cognition of land ecological function	Yes = 1; No = 0	0.4494	0.4978
**Policy Variables**			
Understanding of government’s non-point source pollution control policies	I do not know at all = 1; I know a little = 2; I know roughly = 3; I know very clearly = 4	1.7184	0.8989
Understanding of ecological compensation policies	I do not know at all = 1; I know a little = 2; I know roughly = 3; I know very clearly = 4	1.6899	0.9777

**Table 3 ijerph-17-02484-t003:** Covariance variance of multivariate probit model.

Compensation Method	Capital	Material	Technology	Project
Capital compensation				
Material compensation	−0.0249 (0.0643)			
Technology compensation	−0.2355 *** (0.0644) −0.2678 *** (0.0854)	0.2594 *** (0.0595)		
Project compensation	−0.2774 *** (0.0839)	−0.3884 *** (0.0746)	−0.0716 (0.0817)	
LR test	rho21 = rho31 = rho41 = rho32 = rho42 = rho43 = 0			
chi^2^	chi^2^ (6) = 61.5866			
Prob.	0.0000			

Note: *, ** and *** indicate the significant level of 10%, 5% and 1% respectively.

**Table 4 ijerph-17-02484-t004:** Regression results of multivariate probit model.

Independent Variable	Dependent Variable			
	Capital	Material	Technology	Project
Constant	0.1465 (0.4693)	−0.7162 (0.4398)	−0.1362 (0.4458)	−2.1849 *** (0.5812)
Gender	−0.3830 *** (0.1265)	−0.2180 * (0.1157)	0.1591 (0.1186)	0.3628 ** (0.1651)
Age	0.0062 (0.0059)	0.0248 *** (0.0056)	−0.0042 (0.0055)	−0.0275 *** (0.0073)
Education degree	−0.0075 (0.0162)	0.0187 (0.0152)	0.0507 *** (0.0156)	0.0761 *** (0.0232)
Household income	0.0186 (0.0327)	−0.0558 * (0.0310)	−0.0197 (0.0315)	0.1513 *** (0.0419)
Population burden rate	−0.2522 (0.2256)	−0.1005 (0.2143)	−0.1665 (0.2162)	−0.2557 (0.2926)
Degree of part-time farming	0.2752 *** (0.0688)	−0.0529 (0.0640)	−0.1256 * (0.0655)	−0.1562 * (0.0900)
Cultivated area	−0.0413 *** (0.0125)	−0.0147 (0.0117)	−0.0221 (0.0145)	0.0372 *** (0.0136)
Eco-friendly technology adoption	−0.3002 *** (0.0782)	0.0583 (0.0757)	0.2740 *** (0.0776)	−0.1265 (0.0978)
Risk attitude	0.2525 *** (0.0699)	0.0281 (0.0652)	−0.0833 (0.0676)	0.2013 ** (0.0900)
Cognition of land ecological function	−0.2013 * (0.1147)	0.0671 (0.1091)	0.5307 *** (0.1100)	0.4417 *** (0.1492)
Understanding of government’s non-point source pollution control policies	−0.2542 *** (0.0666)	−0.1741 *** (0.0637)	−0.0530 (0.0649)	0.2275 *** (0.0800)
Understanding of ecological compensation policies	0.0418 (0.0610)	−0.0391 (0.0584)	0.0410 (0.0574)	0.3350 *** (0.0736)
LR	−1335.3851			
Waldchi^2^	Waldchi^2^ (48) = 299.56			
Prob.	0.0000			

Note: *, ** and *** indicate the significant level of 10%, 5% and 1% respectively.

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
