# Peer review of "‘Get a Fish’ vs. ‘Get a Fishing Skill’: Farmers’ Preferred Compensation Methods to Control Agricultural Nonpoint Source Pollution"

_ijerph, 2020, doi:10.3390/ijerph17072484_

Round 1
Reviewer 1 Report
General:
Agricultural nonpoint source pollution (AGNPS) is a critical challenge to water quality protection of many watershed in China. This manuscript tried to characterize the ecological compansation for nonpoint source pollution control. This study is interesting and valuable to the scientific understanding of the relarionship between the nonpoint source pollution control and ecological compansation. However, it might be benefited from additional information on N and P loadings. Meanwhile, authors should pay much attention on representative of the study site and show how important of this study maybe involved in the HanJiang watershed. Authors should also concern what different viewpoint would be concluded from your long term field investigation, compared with short term experiments? What would the optimal distribution of the compansation in the whole watershed scale? I recommend authors to address my above concerns. This article may consider for publication in IJERPH after moderate revision.
Author Response
Response: Thank you for this helpful suggestion.
On the first question, we have rewritten the part in page 1line 34-27, which is “Excessive chemical fertilizers, pesticides, livestock manure in agricultural production have led to N, P, and other pollutants entering the water and the entire ecosystem, manifesting in the form of agricultural nonpoint source pollution”. Besides, we have introduced some information on N and P loadings in page 4 line 5-7 by “For example, owing to water pollution, the Hanjiang River carries 1.7 mg/L of nitrogen when it flows out of Ankang City (Hanzhong City is upstream) [33]”.
On the second question, we have explained the reason of choosing study site on page 4 line3-21, but it may not be clear. This part has been rewritten to make it clearer to the readership, which is:
“There are four main reasons these cities were chosen as the study areas. First, agricultural nonpoint source pollution became the primary source of water pollution in the study area [31-32]. For example, owing to water pollution, the Hanjiang River carries 1.7 mg/L of nitrogen when it flows out of Ankang City (Hanzhong City is upstream) [33], which can be classified into IV types of water according to the China environmental quality standard for surface water. Second, those areas experience excessive fertilizer application. According to the survey data of agricultural production, the inputs of chemical fertilizers for rice, wheat, and rapeseed were 328.35 kg/hm2, 297.15 kg/hm2, and 319.65 kg/hm2, respectively. After considering the characteristics of the double-cropped main crops in the study area (rice, wheat, and rapeseed, respectively), the annual average input of chemical fertilizers per hectare ranged from 625.50 kg to 648.00 kg, which was 2.78-2.88 times the internationally recognized upper limit of fertilizer application (225 kg/hm2). Therefore, the ecological compensation policy in this region holds significance in view of controlling agricultural nonpoint source pollution and ensuring ecological security. Third, these cities are critical water sources for China, so there are many beneficiaries of its agricultural nonpoint source pollution control. They house the headwaters of the Hanjiang River as well as the water conservation areas of the South-to-North Water Transfer Project. Hanjiang River is the largest tributary of the Yangtze River, which is the largest river in China. The South-to-North Water Transfer Project is the largest water conservancy project in China. Once water pollution occurs in this area, the pollutants may spread across the Hanjiang River, even endangering Beijing-Tianjin and the Yangtze River basin. Forth, the control of agricultural nonpoint source pollution in the study area is of great ecological significance. These cities are located in the hinterland of the Qinba Biodiversity Ecological Function Zone, which is a crucial national ecological function area. Hence, measures to control agricultural nonpoint source pollution in the study areas contribute immensely toward the safety of the ecological function area and that of the wildlife habitats. ”
On the third and fourth question, it is necessary to consider the heterogeneity of time and space in the design of agricultural nonpoint source pollution control policies. On the one hand, from the perspective of time, farmers’ individual preferences will transform with the change of environment and policy. However, this paper is the first empirical study on the compensation methods of agricultural nonpoint source pollution control, lacking of comparable research results. On the other hand, from a spatial perspective, discussion of the optimal distribution of compensation between the upper, middle and lower areas of the Hanjiang River Basin is indispensable. But limited to the survey area and data, the optimal distribution of compensation cannot be solved in this paper. Thank you again for your comments and reminders.
According to the reviewer’ suggestion, we have added a discussion section as follows:
“In the follow-up study of compensation, it is necessary to explore temporal and spatial heterogeneity. In the long run, farmers' preference for compensation mode may change with the development of policy, environment and personal ability. Understanding the internal mechanism of this change helps to improve the sustainability and effectiveness of compensation policy. From the spatial dimension, if the costs and benefits of the upper, middle and lower reaches are heterogeneous, differential compensation policies are indispensable. Discussion on how to optimize the distribution of the compensation in the Hanjiang River Basin is an important content of compensation research in the future.”

Reviewer 2 Report
I had no real substantive issues with this paper. I thought the study was well thought-out and well presented.
I did have a questions, however. These minor points are listed below. They are listed by page (but not all pages had issues).
Page 3
Line 26. Double cropped may need a hyphen as it is used as adjective (double-cropped).
Line 26. Extra period at the end of the sentence.
Line 30. The phrase "Concerning the demographics and agricultural statistics ..." to begin that sentence is awkward. The entire phrase is probably unneeded or could be replaced by "With respect to the demographics and agricultural statistics."
Line 35. Similar to what was noted above. In this case, a transition is needed, so the phrase cannot simply be deleted. However, using words like, "As for ..." (rather than Concerning) would make it an easier read.
Page 4
Figure 1 -- Unsure why the county names are not capitalized (treated as proper nouns) on the map. (They are capitalized in the text).
Line 33 -- The high validity rate is impressive. But given that the interviews were conducted face-to-face, it would be instructive (or at least interesting) to know what happened in the 38 cases where the data from the questionnaires was deemed unusable.
Line 37 -- The phrase "trained investigators before the investigation" in confusing. Either eliminate the second half of the phrase (before the investigation) or use a different word so it does not sound so redundant (before the study)>
Page 6
Figure 2 -- Is there a reason for using a coned 3-D bar graph. A regular bar graph, while not as visually interesting, may actually be easier to read.
Page 7
Table 2 -- Does "Education degree (number of years)" refer to the years of formal schooling? I just wanted to confirm as the mean educational attainment level of just over 6 years -- while probably correct in context -- would be viewed as quite low in many settings.
End of page -- The preference selections appear to almost be mutually exclusive. This may be something to note in the discussion of the analysis.
Page 9
Line 11 -- Are the farmers who select GFSM more economically or environmentally conscious? The text states economically but in some ways it would seem that they would be more concerned with environmental matters. Unless this is foreshadowing the discussion that follows where the move to technology was seen as a way to improve productivity and this profitability which is discussed in the next paragraph. (In other words, profits and not the protecting the planet are why these farmers seek technology and thus prefer GFSM). This might be a point worth expanding upon.
Author Response
Page 3
Line 26. Double cropped may need a hyphen as it is used as adjective (double-cropped).
Response: Thank you for this helpful suggestion, we are very sorry for this typo and have corrected it.
Line 26. Extra period at the end of the sentence.
Response: Thank you for this helpful suggestion, we apologize for this typo and have corrected it.
Line 30. The phrase "Concerning the demographics and agricultural statistics ..." to begin that sentence is awkward. The entire phrase is probably unneeded or could be replaced by "With respect to the demographics and agricultural statistics."
Response: Thank you for this helpful suggestion, we have changed "Concerning the demographics and agricultural statistics ..." into "With respect to the demographics and agricultural statistics ..."
Line 35. Similar to what was noted above. In this case, a transition is needed, so the phrase cannot simply be deleted. However, using words like, "As for ..." (rather than Concerning) would make it an easier read.
Response: Thank you for this helpful suggestion, we have changed “Concerning” into "As for ..."
Page 4
Figure 1 -- Unsure why the county names are not capitalized (treated as proper nouns) on the map. (They are capitalized in the text).
Response: Thank you for this helpful suggestion, we have capitalized the county names.
Line 33 -- The high validity rate is impressive. But given that the interviews were conducted face-to-face, it would be instructive (or at least interesting) to know what happened in the 38 cases where the data from the questionnaires was deemed unusable.
Response: The 38 cases may be the status that some farmers are reluctant to provide personal information, or walk away without completing the whole questionnaire. These situations are unavoidable in the questionnaire survey.
Line 37 -- The phrase "trained investigators before the investigation" in confusing. Either eliminate the second half of the phrase (before the investigation) or use a different word so it does not sound so redundant (before the study)>
Response: Thank you for this helpful suggestion, “before the investigation” has been omitted in the revised manuscript.
Page 6
Figure 2 -- Is there a reason for using a coned 3-D bar graph. A regular bar graph, while not as visually interesting, may actually be easier to read.
Response: Thank you for this helpful suggestion, we have changed Figure 2 into regular bar graph.
Page 7
Table 2 -- Does "Education degree (number of years)" refer to the years of formal schooling? I just wanted to confirm as the mean educational attainment level of just over 6 years -- while probably correct in context -- would be viewed as quite low in many settings.
Response: Yes, "Education degree (number of years)" refers to the years of formal schooling. This is a common data in rural China, it conforms to the statistical data of China Rural Statistical Yearbook and Shaanxi Provincial Statistical Yearbook.
End of page -- The preference selections appear to almost be mutually exclusive. This may be something to note in the discussion of the analysis.
Response: Thank you for this helpful suggestion, we have deleted “and less likely to choose GFSM”.
Page 9
Line 11 -- Are the farmers who select GFSM more economically or environmentally conscious? The text states economically but in some ways it would seem that they would be more concerned with environmental matters. Unless this is foreshadowing the discussion that follows where the move to technology was seen as a way to improve productivity and this profitability which is discussed in the next paragraph. (In other words, profits and not the protecting the planet are why these farmers seek technology and thus prefer GFSM). This might be a point worth expanding upon.
Response: Thank you for this helpful suggestion. The reviewer’ suggestion is quite enlightening. The reason why farmers choose GFSM may be simply because of profits, or simply for the purpose of environmentally, more likely both. It is a complex problem to explain this reason. We need to do psychoanalysis, which is one of our follow-up research contents. Thank you for your comment again.
Reviewer 3 Report
The paper investigates the farmers' response to farm control policies aimed to control water quality affected by non-point pollution sources.
The manuscript is well organized, the reader easily reads the text and figures and the subject matter is of certain relevance.
Hence, the paper can be judged to add knowledge in its topic and it adheres to the “Int. J. Environ. Res. Public Health” standards,
The article can therefore be published in its current form, except for the need for a more in-depth discussion of the results, probably in the specific "Discussion" section.
For example: what can planners do to increase the orientation of farmers towards the more complex compensation modes of GFSM?
Furthermore: “biological pesticides” is nota n appropriate term in agriculture.
Author Response
Response: Thank you for this helpful suggestion. We are very sorry for this typo and have corrected “biological pesticides” into “microbial pesticides”
Besides,according to the reviewer’ suggestion, we have added a discussion section as follows:
“In the process of promoting GFSM, the government should improve farmers' acceptance, at least, from the following two aspects. On the one hand, poblicity can improve farmers' awareness that GFSM has its sustainable and economic advantages. At the same time, technical training of environmental-friendly protection production can improve farmers' ability to participate in GFSM. On the other hand, local economic conditions, employment of farmers, and local resources should be well considered during the process of specific GFSM design and implementation to ensure the compensation implement successfully.”
